



**Mechanisms and impacts of extreme high-salinity shelf water formation in the Ross Sea**
**Xiaoqiao Wang[1, 2] Zhaoru Zhang[3*, 4, 5, 6] Chuan Xie[3] Xi Zhao[7 8] Chuning Wang[3, 4, 5] Heng Hu[3] Yuanjie**
**Chen[3]**
[1] College of Meteorology and Oceanography, National University of Defense Technology, Changsha,
China
[2] Key Laboratory of High Impact Weather(special), China Meteorological Administration, Beijing, China
[3] Key Laboratory of Polar Ecosystem and Climate Change,Ministry of Education and School of
Oceanography, Shanghai Jiao Tong University, 1954 Huashan Road, Shanghai, 200030, China
[4] Shanghai Key Laboratory of Polar Life and Environment Sciences, Shanghai Jiao Tong University,
Shanghai, China
[5] Shanghai Frontiers Science Center of Polar Science, Shanghai Jiao Tong University, 1954 Huashan Road,
Shanghai, 200030, China
[6] Key Laboratory for Polar Science, Polar Research Institute of China, Ministry of Natural Resources,
Shanghai, 200136, China
[7] School of Geospatial Engineering and Science, Sun Yat-sen University, and Southern Marine Science and
Engineering Guangdong Laboratory (Zhuhai), Zhuhai, 519000, China
[8] Key Laboratory of Comprehensive Observation of Polar Environment (Sun Yat-sen University), Ministry
of Education, Zhuhai, 519082, China
Corresponding author: Zhaoru Zhang (zrzhang@sjtu.edu.cn)



**Abstract**
High-salinity shelf water (HSSW) acts as a precursor to the Antarctic Bottom Water (AABW) and plays a
critical role in regulating the global ocean circulation system. This study employs a high-resolution coupled
ocean-sea ice-ice shelf model to analyze the interannual variation in HSSW formation in the Ross Sea,
which is one of the major production sites of HSSW. We are particularly focused on anomalously high
HSSW production during the winter of 2007. The results indicate that in this winter, there were frequent
passages of synoptic-scale cyclones that are centered near the front of the Ross Ice Shelf. The western
flanks of these cyclones significantly enhanced offshore winds over the western Ross Ice Shelf polynya, a
major origin site of HSSW in the Ross Sea, leading to a sharp increase in ice production within this polynya.
The HSSW formation resulting from brine rejection during ice freezing reached the highest volume of
16,000 km³ in 2007. However, salinity and density of the Ross Sea during this period exhibited unexpected
low values. Such inconsistency was due to a rapid increase in ice shelf melting over the Amundsen Sea and
Ross Seas during 2006−2007, with annual cumulative melt rates reaching the peak in recent decades.
Meanwhile, the resulting large amount of meltwater was transported westward into the Ross Sea by notably
strong slope and coastal currents in 2007, leading to large fluxes of freshwater flux into the Ross Sea. The
interaction between enhanced HSSW formation driven by ice production and the large influx of meltwater
highlights the complex dynamics that shape hydrographic variability in the Ross Sea.



## 1 Introduction

High-salinity shelf water (HSSW), a precursor to the Antarctic Bottom Water (AABW), is predominantly formed in Antarctic coastal polynyas, which are regions of persistent open water bordered by sea ice along the coastline. These polynyas are largely driven by katabatic and synoptic offshore winds (Bromwich et al., 1998; Massom et al., 1998; Morales Maqueda et al., 2004; Weber et al. 2016; Wenta and Cassano, 2020), which enhance air-sea heat exchange, expand polynya size, and increase sea ice production (SIP) as wind speed rises. During the freezing seasons, the continuous sea ice formation and associated brine rejection in coastal polynyas lead to the formation of HSSW. When HSSW mixes with relatively warmer water masses, such as the Circumpolar Deep Water (CDW) and ice shelf meltwater (ISW), the resulting water mass can cross the continental slope and sink to the deep ocean, ultimately forming AABW (Comiso and Gordon, 1998; Ohshima et al., 2013; Whitworth et al., 2013). As the lower limb of the global overturning circulation, AABW plays a crucial role in regulating the oceanic heat storage capacity and the pathways of carbon sequestration in the Southern Ocean and the world ocean (Arrigo et al., 2008; Gruber et al., 2019; Murakami et al., 2020; Li et al., 2023).

The Ross Sea (Fig. 1a) is a primary region for the formation of AABW (Gordon et al., 2009), with a production rate accounting for approximately 20–40% of the total (Meredith, 2013; Solodoch et al., 2022). AABW in the Ross Sea primarily originates from two typical coastal polynyas: the Terra Nova Bay Polynya (TNBP) in the western Ross Sea and the Ross Ice Shelf Polynya (RISP) (Fig. 1b); the latter has the highest ice production among all Antarctic coastal polynyas (Tamura et al., 2016) and is the focused region of this study. The western side of RISP serves as the primary site for HSSW formation, while the eastern side, significantly influenced by the fresh ISW, is less favorable for the formation of high-salinity HSSW (Smith et al., 2012; Yan et al., 2023). Previous studies revealed the formation and characteristics of HSSW and AABW in the Ross Sea on synoptic to interannual time scales as well as long-term trends, demonstrating the roles of atmospheric circulations, sea ice production and freshwater input. At the synoptic scale, strong wind events driven by cyclones or katabatic winds play a critical role in shaping the polynya dynamics and extent. Numerous studies have identified strong correlations between wind speed and sea ice concentration/production in the polynyas (Bromwich et al., 1998; Dale et al., 2017; Cheng et al., 2019; Ding et al., 2020; Wenta and Cassano, 2020); the variability of sea ice production further influences ocean convection, the HSSW formation (Thompson et al., 2020; Wang et al., 2021) and its subsequent transport (Wang et al., 2023). On the seasonal scale, based on mooring datasets and numerical simulations, significant HSSW production starts in July and lasts until October (Mathiot et al., 2012; Rusciano et al.,





2013; Yan et al., 2023). For the long-term trend, observations show a dramatic decline in the HSSW salinity
over recent decades, attributed to increased transport of ISW from the Amundsen Sea to the Ross Sea
(Jacobs et al., 2022), though such trend has been found to be reversed in recent years (Castagno et al., 2019;
Silvano et al., 2020; Guo et al., 2021). Up until now, there have been no studies focusing on extreme HSSW
production events, which can make significant contributions to the HSSW volume and subsequently the
AABW production. In this study, by examining the distinct variations in HSSW formation from 2003 to
2019, we aim to elucidate the underlying physical processes driving anomalously high HSSW production
in the austral winter of 2007. A high-resolution coupled ocean-sea ice-ice shelf model covering the Ross
Sea and the Amundsen Sea, which effectively simulates the observed temporal variability of HSSW, is
employed to perform such investigations. Bottom salinity and density fields in the Ross Sea are also
analyzed, the variations of which suggest the combined effects of HSSW production and freshwater input.
The manuscript is organized as follows: Section 2 describes the numerical model, observational data, model
validation, and analysis methods. Section 3 presents results on the atmospheric drivers of SIP and their
influence on HSSW formation, as well as the impacts of meltwater transport from the Amundsen Sea.
Section 4 discusses the relationships between large-scale atmospheric modes and HSSW variability.
Section 5 provides the conclusions.

## 91  2 Date and Methods

2.1 Model data description
This study utilizes a high-resolution Ross-Amundsen Sea ocean-sea ice-ice shelf model (RAISE, Zhang et
al., 2024b) developed based on the Regional Ocean Modeling System (ROMS v3.6), a primitive-equation,
free-surface, terrain-following coordinate model (Shchepetkin and McWilliams, 2009). ROMS is coupled
with a dynamic sea ice model (Budgell, 2005) employing elastic-viscous-plastic (EVP) rheology (Hunke
and Dukowicz, 1997; Hunke, 2001) which includes two-layer ice thermodynamics with snow as an
insulating layer following Mellor and Kantha (1989) and Häkkinen and Mellor (1992). This configuration
effectively simulates sea ice characteristics in polar regions around the Antarctic including the Ross Sea
(Stern et al., 2013; Dinniman et al., 2011, 2015). The ice shelves are modeled as static, without mass
variation or iceberg calving, and a three-equation parameterization scheme is employed to represent the
thermodynamic and mechanical effects of the ice shelf-water interactions (Holland and Jenkins, 1999;
Dinniman et al., 2011).





104 The model domain spans approximately 85.6°S to 64.2°S and 143.0°E to 89.9°W (Fig. 1a), including the

105 Ross Sea and Amundsen Sea, along with the floating ice shelves. The horizontal resolution varies from 2–

106 4 km along the continental shelf to 3–6 km in the open ocean, permitting mesoscale eddies on the

107 continental shelf and slope but is not fully eddy-resolving (Hallberg, 2013; Stewart and Thompson, 2015;

108 St-Laurent et al., 2013). In the vertical dimension, the model comprises 32 terrain-following levels, which

109 provide higher resolution data in the near-surface and bottom layers. The bathymetry and ice shelf

110 topography are derived from MEaSUREs BedMachine Antarctica, Version 2 (Morlighem et al., 2020).

111 ROMS calculates the momentum, heat, and freshwater (imposed as a salt flux) fluxes in the open ocean

112 using the COARE version 3.0 bulk flux formulae (Fairall et al., 2003). The vertical momentum and mixing

113 are calculated using the K-profile parameterization scheme (Large et al., 1994). The initial temperature and

114 salinity conditions derive from a 10-km circum-Antarctic ocean–sea ice–ice shelf model (Dinniman et al.,

115 2015). Boundary conditions for temperature, salinity, sea surface height, and depth-averaged velocities are

116 sourced from the Met Office Global Seasonal Forecasting System version 5 (GloSea5) (Maclachlan et al.,

117 2015), while sea ice concentration data is incorporated from multiple satellite products including the

118 Advanced Microwave Scanning Radiometer-Earth Observing System (AMSR-E), Special Sensor

119 Microwave Imager/Sounder (SMMI/S) and Advanced Microwave Scanning Radiometer 2 (AMSR-2)

120 based on their availability during different time periods. Tidal forcing, based on TPXO-9, includes 15 major

121 tidal constituents (Egbert and Erofeeva, 2002) and is applied at the open boundaries through sea surface

122 height and barotropic currents. Atmospheric forcing fields utilized in this model are obtained from the

123 ERA5 reanalysis product, including 3-hourly surface wind and air temperature, along with daily sea level

124 pressure, precipitation, humidity and cloud cover (Hersbach et al., 2020) produced by the European Centre

125 for Medium-Range Weather Forecasts (ECMWF). The model simulation spans from 2003 to 2019 after a

126 5-yr spin-up simulation, and the model results are output as 5-day-average fields.

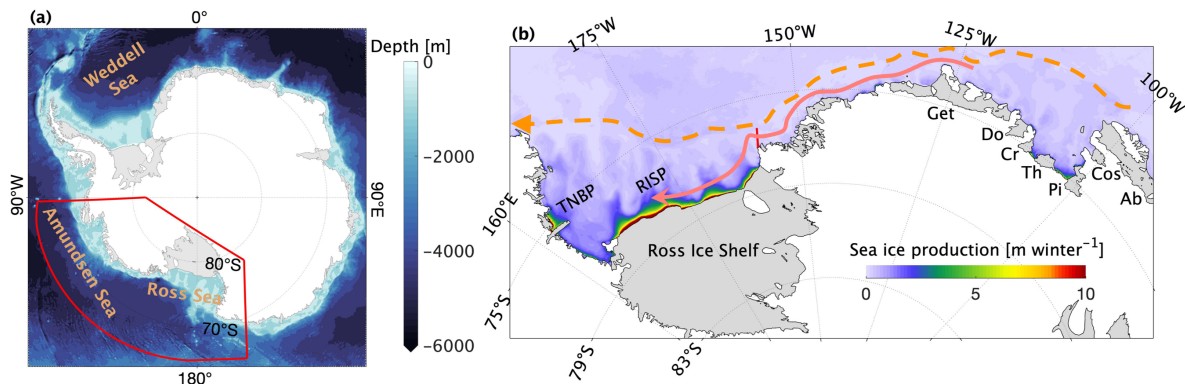

127



**Figure 1.** Geographic maps of **(a)** the Southern Ocean south of 60°S and **(b)** the Ross Sea and Amundsen Sea. Areas in white show continental surfaces, and areas in light grey indicate ice shelves. The color scale indicates accumulated sea ice production over the austral winter, averaged from 2003–2019 based on the model simulations. In panel (a), the model domain is shown by the solid red box. In panel (b), the dashed yellow line indicates the Antarctic slope current, the solid pink line indicates the coastal current, and the red line between the Ross Sea and the Amundsen Sea indicates the selected transect used to calculate the freshwater transport. RISP represents the Ross Ice Shelf Polynya and TNBP indicates the Terra Nova Bay Polynya. The labeled local ice shelves are: Abbot Ice Shelf (Ab), Cosgrove Ice Shelf (Cos), Pine Island Ice Shelf (Pi), Thwaites Ice Shelf (Th), Crosson Ice Shelf (Cr), Dotson Ice Shelf (Do), and Getz Ice Shelf (Get).

2.2 Model validation

A comprehensive validation of the RAISE model is presented by Zhang et al. (2024b), demonstrating its capacity to effectively capture the temporal and spatial variability of sea ice production in polynyas and the hydrographic properties of HSSW in the Ross Sea. With regard to sea ice area (SIA), a comparison with the AMSR-E/AMSR2 datasets indicates that the simulated temporal variability of SIA is highly correlated with that of observed dataset (R = 0.91, P < 0.001), though SIA is typically underestimated in the model. The model effectively captures the interannual variations of accumulative SIP in the Ross Sea polynyas, with correlations between the modeled and satellite-retrieved SIP reaching 0.62 for the Terra Nova By polynya (Zhang et al., 2024b) and 0.64 for the western RISP (i.e. the major HSSW formation site over the entire RISP, Fig. 2) at the 95% confidence level. The satellite-retrieved SIP dataset is available from the Institute of Low Temperature Science at Hokkaido University (http://www.lowtem.hokudai.ac.jp/wwwod/polar-seaflux/southern_ocean_new/AMSR-POLAR/).

The performance of the RAISE model in simulating the hydrographic properties and water masses in the Ross Sea was evaluated using temperature-salinity (T-S) diagrams and vertical transect patterns from the World Ocean Database (WOD) and the Marine Mammals Exploring the Oceans Pole to Pole (MEOP) seal-tag CTD observations (Zhang et al. 2024b). The model accurately reproduces the key water masses present in the region, including the HSSW, CDW, and ISW. Furthermore, the model exhibits a high degree of accuracy in reproducing the temporal variability of HSSW density and salinity over the Ross Sea. A comparison of the model output with mooring data from the Italian MORSea and U.S. CALM projects revealed significant correlations between the variations of modeled and observed HSSW density in the Terra Nova Bay polynya and on the western Ross Sea slope. More details of the model validation are





referred to Zhang et al. (2024b). Meanwhile, the simulated ice shelf melt rates for the Amundsen Sea and
Ross Sea show good agreement with observation-based estimates (Xie et al., 2024).

2.3 Methods
The HSSW is defined as the water mass with neutral density $(\gamma^n) > 28.27$ kg m$^{-3}$, practical salinity $(S) >$
34.62 and potential temperature $(\theta) <$ -1.85°C (Orsi and Wiederwohl, 2009; Castagno et al., 2019). The
area for calculating the HSSW volume extends to the base of the Ross Ice Shelf, as Assmann et al. (2003)
and Budillon et al. (2003) demonstrated the existence of a southerly flow on the western side of the Ross
Sea, transporting HSSW to the base of the ice shelf. This flow can advect more than 10% of HSSW to the
southern part of RIS and intensify continuously over winter (Jendersie et al., 2018).

Cyclone tracking was conducted using the University of Melbourne Automatic Cyclone Tracking Scheme
(Murray and Simmonds, 1991), which is based on the ERA5 reanalysis product from 2003 to 2019. The
optimal parameters for this scheme, including the horizontal air pressure field smoothing parameter, the
radius for calculating Laplacian pressure, and the maximum topographic height employed for cyclone
detection, were derived from the values established by Uotila et al. (2009). The identified cyclones were
characterized by a number of properties, including their locations, lifetimes, mean radii, and additional
characteristics. Cyclones were selected based on specific criteria: a lifetime exceeding 12 hours and a
distance greater than 1,000 km between the first and last detected locations. Such criteria can exclude
certain detected but unrealistic cyclones (Uotila et al., 2011). The region south of 42°S was divided into
720 sectors, each encompassing 4° in latitude and 6° in longitude. The number of cyclone tracks per sector
was calculated for the period from 2003 to 2019, following the definitions proposed by Uotila et al. (2013).

The transport of meltwater discharged from ice shelves is quantified by calculating the specific freshwater
transport across a designated meridional transect at the boundary between the Ross and Amundsen Seas
(Figure 1b). Following Li et al. (2021), the advective freshwater transport across the boundaries is
calculated as follows:

$$Q_{fw} = \iint \boldsymbol{u} F^{fw} dz dl, \tag{1}$$

where $\boldsymbol{u}$ represents the horizontal velocity vector (u, v), and $F^{fw}$ is freshwater content. $F^{fw}$ is derived from
Brown and Irish (1993) as follows:



$$F^{fw} = \frac{S_{ref} - S}{S_{ref}}, \qquad (2)$$

with $S$ as the salinity and $S_{ref}$ set to 34.9 PSU as a reference, which is the maximum salinity observed in the Ross Sea. This approach quantifies the freshwater input by combining spatially integrated velocity and salinity contrasts, offering insights into the transport dynamics of meltwater across defined oceanic boundaries.

## 3 Results

3.1 Sea ice production and the related atmospheric drivers

Sea ice production in the coastal polynya is the determinate factor for HSSW formation. From 2003 to 2019, modeled cumulative ice production over the RISP during austral winter exhibited a pronounced peak in 2007 (Fig. 2). Satellite-derived ice production data from 2003 to 2013 also show the maximum value in 2007.

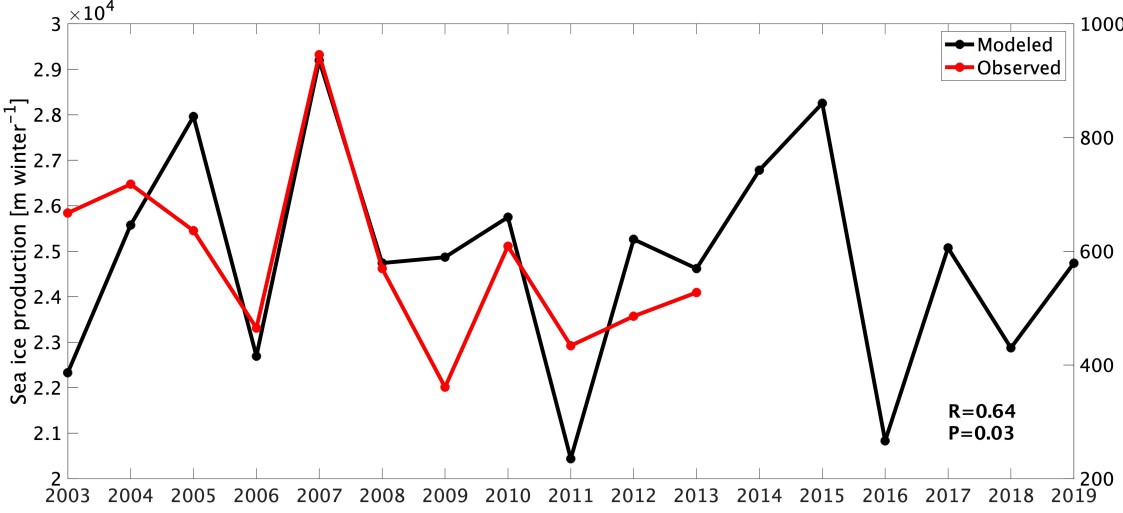

**Figure 2.** Time series of observed (red line; 2003–2013) and modeled (black line; 2003–2019) polynya-averaged sea ice production (SIP) during austral winter (June–August) for the western Ross Ice Shelf Polynya (left y-axis: modeled SIP, right y-axis: observed SIP). The correlation coefficient between the observed and modeled SIP is 0.64 (P = 0.03).



To better understand the drivers behind the exceptionally high ice production in 2007, statistical analysis of cyclones over the Ross Sea and surrounding regions was performed using the cyclone tracking scheme. The spatial distribution of winter cyclone track density (Fig. 3) reveals that most cyclone centers are concentrated north and east of the Ross Sea, with relatively few or no cyclones observed over the Ross Ice Shelf, particularly between 79° and 84°S. However, in several years such as 2005, 2007, 2017 and 2019, relatively high frequency of cyclones was recorded over the Ross Ice Shelf (Figs. 3c, e, o, q). Among these years, 2007 stands out as the year with the highest cumulative cyclone track density, reaching approximately 18 months (Fig. 3e).

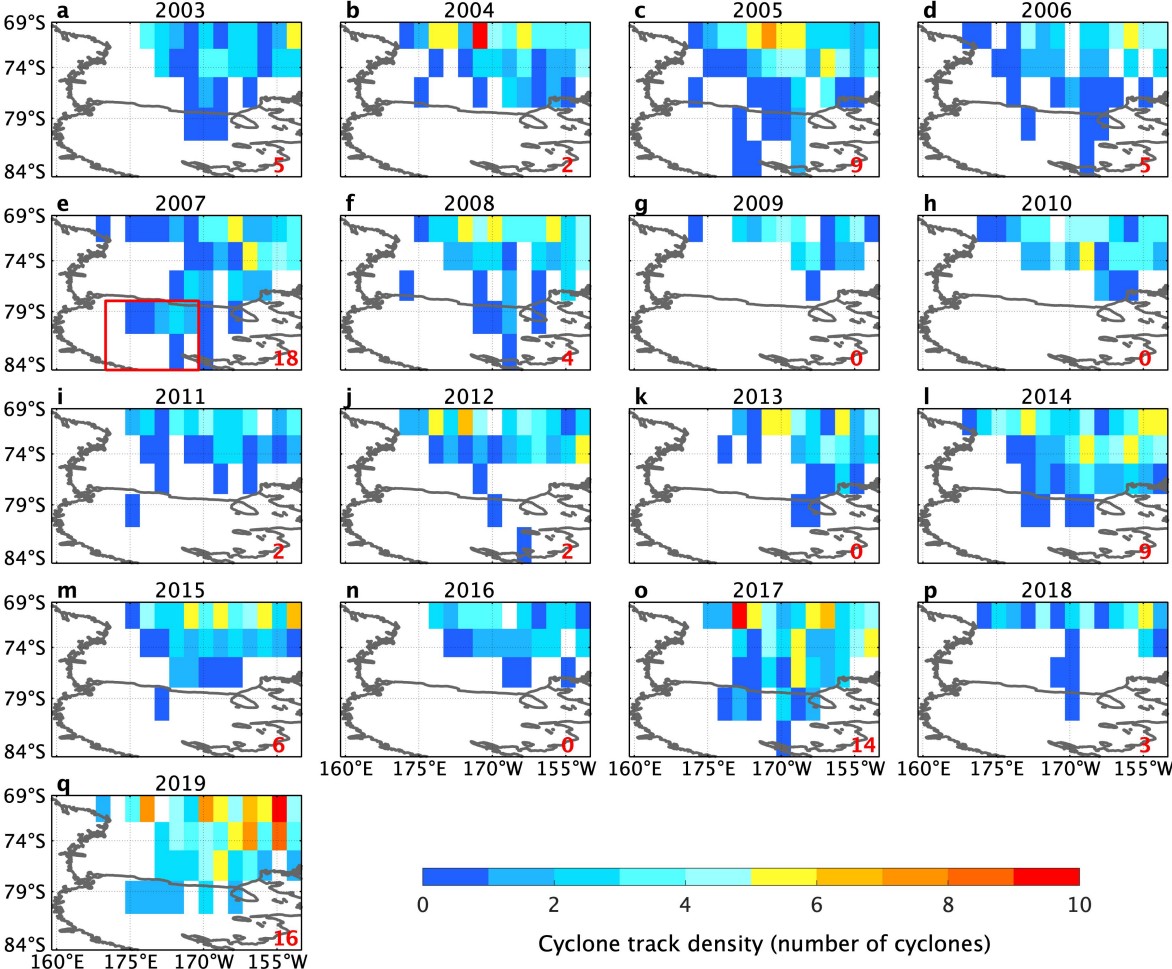

**Figure 3. (a–q)** Austral winter track densities (the number of tracks per section) of cyclones in the Ross Sea from 2003–2019. The red box on the Ross Ice Shelf in (e) represents the spatial extent over which the





accumulated track density is calculated, and the numbers in the lower right corner indicate the density value
calculated based on this box.

The frequent passage of cyclones centered near and over the Ross Ice Shelf significantly impacts the local
wind field. The western branches of these cyclones enhance offshore winds over the western RISP, resulting
in intensified meridional wind speeds (Fig. 4). In 2007, the aggregation of cyclones in this region (Fig. 3e)
led to pronounced increases in offshore winds (Fig. 4e). The spatial distribution of meridional wind speed
for multiple years further highlights that, in 2007, offshore winds over the western RISP were the strongest,
reaching approximately 9 m s$^{-1}$ (Fig. 4e). Additionally, the area with offshore wind speeds exceeding 5 m
s$^{-1}$ extended farther eastward compared to other years, reaching approximately 175°W (Fig. 4e). Fig. 4r
highlights that the spatially averaged meridional wind speed on the western side of the Ross Sea (red box
shown in Fig. 4e) in 2007 was approximately 6.5 m s$^{-1}$, the highest among the analyzed years. Additionally,
other years with elevated cyclone track densities, such as 2005, 2017, and 2019 (Figs. 3c, o, q) also
exhibited relatively strong offshore winds (Figs. 4c, o, q), with average wind nearing 6 m s$^{-1}$ (Fig. 4r). This
intensification of offshore winds in winter of 2007 drives the record-high SIP in the RISP during this year
(Fig. 2). Similarly, the relatively strong offshore wind speeds in 2005, 2017 and 2019 also contributed to
notable increases in SIP in these years (Fig. 2).







**Figure 4. (a–q)** Spatial distributions of winter-averaged 10-m meridional wind speed (colored shading) and 10 m wind vectors (black arrows) in the Ross Sea over 2003–2019. **(r)** Time series of spatially averaged meridional wind speed over the western Ross Sea (red box in panel e).

## 3.2 Hydrographic properties and HSSW formation

The brine rejection process associated with sea ice formation in polynya regions directly affects the salinity of the surrounding area. To understand the influence of SIP on the hydrographic characteristics of the Ross Sea, the spatial distribution of winter-averaged salinity from 2003 to 2019 was analyzed firstly (Fig. 5). Typically, regions of high SIP, such as the RISP, are expected to exhibit higher salinity levels due to



enhanced brine rejection. However, contrary to the expectations, salinity values in the western Ross Sea in
2007 were not high, ranging between 34.5–34.7 (Fig. 5e). These salinity levels were lower than those
recorded in other years with moderate SIP, such as 2005, 2017 and 2019, where salinity reached 34.8–34.9
(Fig. 5c, o, q). Similarly, hydrographic properties representing water masses, such as potential density and
neutral density, also showed lower values in 2007 (not shown). These findings indicate that the
hydrographic characteristics in 2007 do not align with the anticipated impacts of the highest SIP occurring
in this year, suggesting the presence of other factors influencing the salinity and density in the Ross Sea.
These processes will be discussed in Section 3.3.

**Figure 5. (a–q)** Spatial distributions of depth-averaged winter salinity in the Ross Sea over 2003–2019.



Salinity is not a precise indicator for quantifying the volume of HSSW formed, therefore the following analysis is instead focused on changes in HSSW volume itself, which can better represent the net production of HSSW. The HSSW volume change in the Ross Sea from 2003 to 2019 is shown in Fig. 6. As mentioned in Section 2.3, the calculation included the region beneath the ice shelf, as indicated by the dashed box in Fig. 6a, where a southward flow on the western side of the Ross Sea can facilitate the transport of newly formed HSSW (Assmann et al., 2003; Budillon et al., 2003; Jendersie et al., 2018). The results revealed that the largest increase in the HSSW volume occurred in the winter of 2007, reaching approximately 16,000 km³ (Fig. 6b). This substantial increase is consistent with the exceptionally high SIP observed in 2007 over RISP (Fig. 2), indicating that significant HSSW formation was triggered by the enhanced brine rejection resulting from ice production in this year. Additionally, the SIP over the TNBP and the corresponding HSSW production in its surrounding region during 2007 were examined. The results revealed that the SIP and HSSW volume increase in 2007 were moderate in this area over 2003 to 2019 (not shown), suggesting that the greatest HSSW volume increase in 2007 in the Ross Sea was primarily driven by the significantly enhanced ice production over RISP. The second-highest increase in HSSW volume was recorded in 2019, which can also be attributed to the relatively strong meridional winds (Fig. 4q) that drove enhanced ice production and HSSW formation. In conclusion, while 2007 experienced the highest SIP and HSSW production, the expected increase in salinity and density was not observed, suggesting complex interactions between SIP and hydrographic properties in the Ross Sea. In the following section, the processes affecting the relationship between the production of sea ice and HSSW and salinity in the Ross Sea in 2007 will be analyzed.

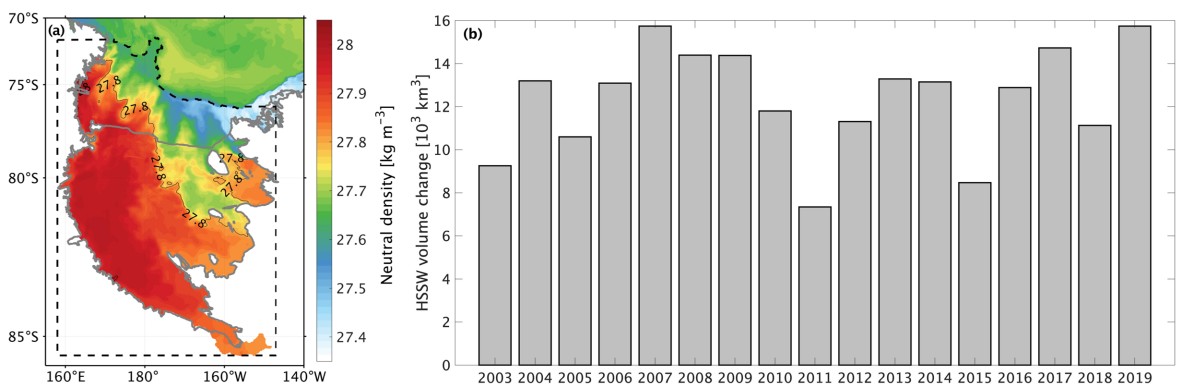

**Figure 6. (a)** Spatial distribution of neutral density (colored shading) in the Ross Sea and Ross Ice Shelf. The solid grey line indicates the neutral density contour of 27.8 kg m⁻³. The dashed black line indicates the



extent of HSSW calculated for HSSW volume; **(b)** Time series of HSSW volume change between October
and July over 2003 to 2019.

3.3 Ice shelf melt water fluxes
In addition to HSSW, the hydrographic characteristics of the Ross Sea are influenced by other water masses
with different properties, such as CDW from the open ocean and ISW. CDW is characterized by relatively
high salinity and temperature (Yabuki et al., 2006; Liu et al., 2017; Morrison et al., 2020; Chen et al., 2023),
which suggests that the most direct factor affecting the anomalously low salinity signal in 2007 is ISW.
Therefore, the winter-averaged ISW volume in the Ross Sea was calculated, identifying a peak in 2007,
with a volume of approximately $3.2 \times 10^3$ km³ (Fig. 7a). This fact indicates that the anomalously low salinity
signal in 2007 (Fig. 5e) was associated with an increase in ISW during that year. The ISW in the Ross Sea
primarily originates from two sources: meltwater from the Amundsen Sea ice shelves and the Ross Ice
Shelf itself, with the former mainly transported to the Ross Sea by the Antarctic slope and coastal currents
(Dinniman et al., 2016; Kusahara and Hasumi, 2013, 2014; Nakayama et al., 2014, 2020; Xie et al., 2024).
The annual cumulative melt rates of both the Ross and Amundsen Sea ice shelves were quantified (Fig. 7b),
and it was found that ice shelf melting intensified beginning in 2006 and remained elevated throughout
2007. This considerable increase in meltwater resulted in large release of fresh ice shelf meltwater, with
the cumulative melt rate in 2007 reaching its highest value observed in recent decades (Fig. 7b).
Furthermore, a comparison of the melt rates of the Ross Ice Shelf and the Amundsen Sea ice shelves
indicates that the magnitude of melting of the latter is significantly greater than that of the former, which is
consistent with the findings of previous studies (Rignot et al., 2013). These results indicate that the
anomalously low salinity observed in the western Ross Sea in 2007 is associated with an increased influx
of ISW, driven by enhanced melting of ice shelves in the Amundsen Sea.



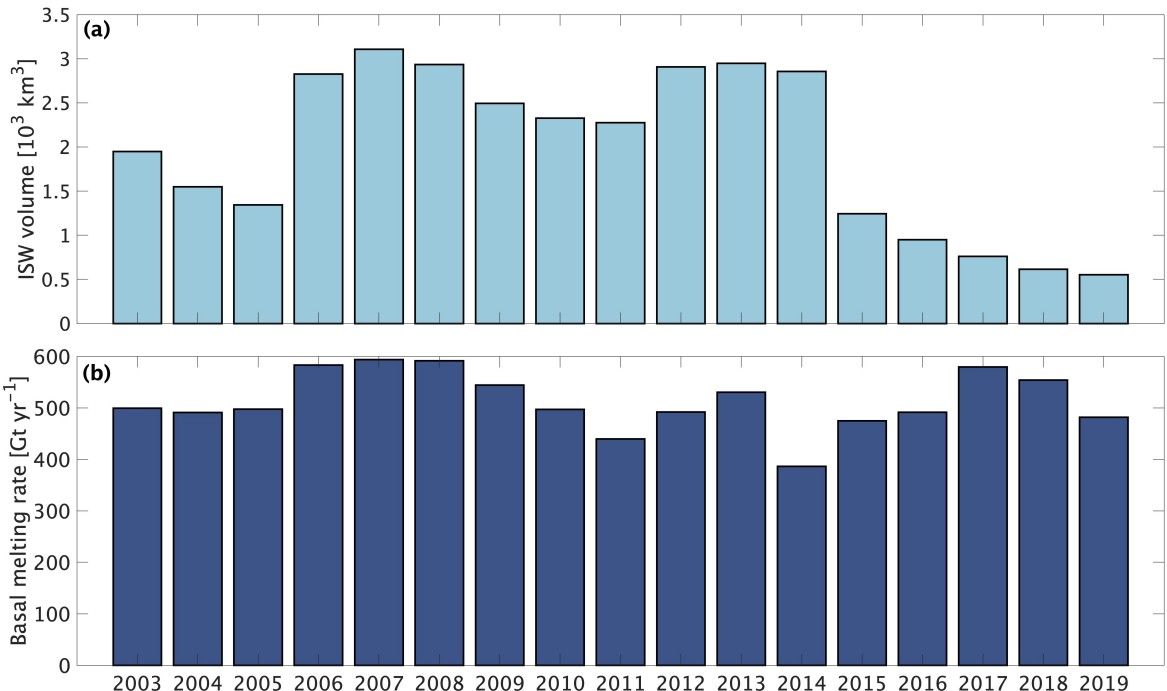

**Figure 7.** Time series of **(a)** winter-averaged Ice Shelf Meltwater (ISW) volume in the Ross Sea and **(b)** annual cumulative basal melting rate for the Amundsen Sea ice shelves and Ross Ice Shelf over 2003 to 2019.

To further demonstrate the impact of Amundsen Sea ice shelf meltwater on the hydrographic properties of the Ross Sea, a meridional transect along the boundary between the Ross Sea and the Amundsen Sea was selected (red line in Fig. 1b). The seasonal mean freshwater fluxes across this transect for all four seasons (austral summer, autumn, winter, and spring) from 2003 to 2019 were calculated (Fig. 8). The time series of seasonal mean freshwater flux during winter (red line in Fig. 8) demonstrates that the absolute value of freshwater flux peaked in 2007, reaching approximately $-5.8 \times 10^4$ Sv. This indicates a substantial westward transport of ISW from the Amundsen Sea into the Ross Sea, consistent with the peak ice shelf melt rate observed in 2007 (Fig. 7b). In addition, it is found that the ISW volume during winter was also highest in 2007, confirming the fact that significant amount of ISW accumulated during this year (Fig. 7a). A similarly strong westward ISW flux was observed during autumn 2007 (black line in Fig. 8). The summer (December to February) freshwater flux shows the second most negative value in 2007, which was $-3.8 \times 10^4$ Sv, indicating a notable westward transport of meltwater into the Ross Sea in late 2006 and early 2007. Spring (September–November) freshwater flux further shows that a significant westward transport was already

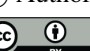


present during spring 2006, while in winter 2006 the transport was weak. These results suggest that the
strong westward ISW flux began in late 2006 and continued to intensify into the autumn and winter of 2007.
The anomalously low salinity values observed in the western Ross Sea in winter 2007 (Fig. 5e) suggest that
the time required for ISW to be transported from the western Amundsen Sea to the western Ross Sea and
significantly impact the local hydrographic properties is approximately 9–12 months. This transport of ISW
played a key role in influencing the hydrographic conditions of the western Ross Sea, particularly
contributing to the unusually low salinity observed in winter 2007 (Fig. 5e). This emphasizes the
importance of understanding how the timing and persistence of ISW fluxes affect the downstream
hydrographic properties in the Ross Sea.

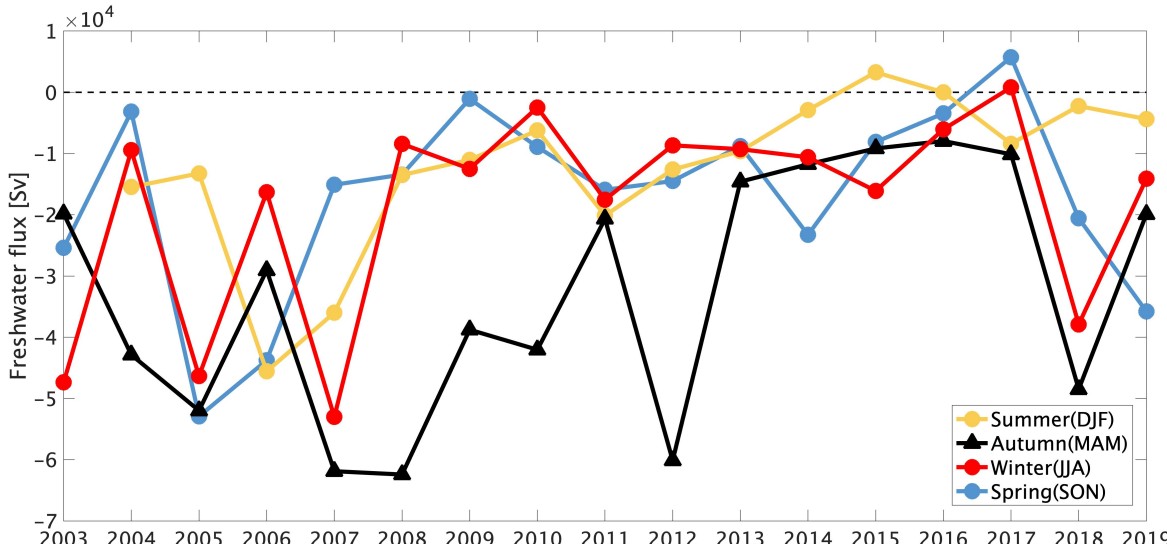


**Figure 8.** Time series of seasonal mean freshwater flux across the defined transect (indicated by the red
line in Fig. 1b) from 2003 to 2019 for austral summer (DJF, yellow line), autumn (MAM, black line), winter
(JJA, red line) and spring (SON, blue line). Negative values represent westward transport of freshwater
from the Amundsen Sea into the Ross Sea.





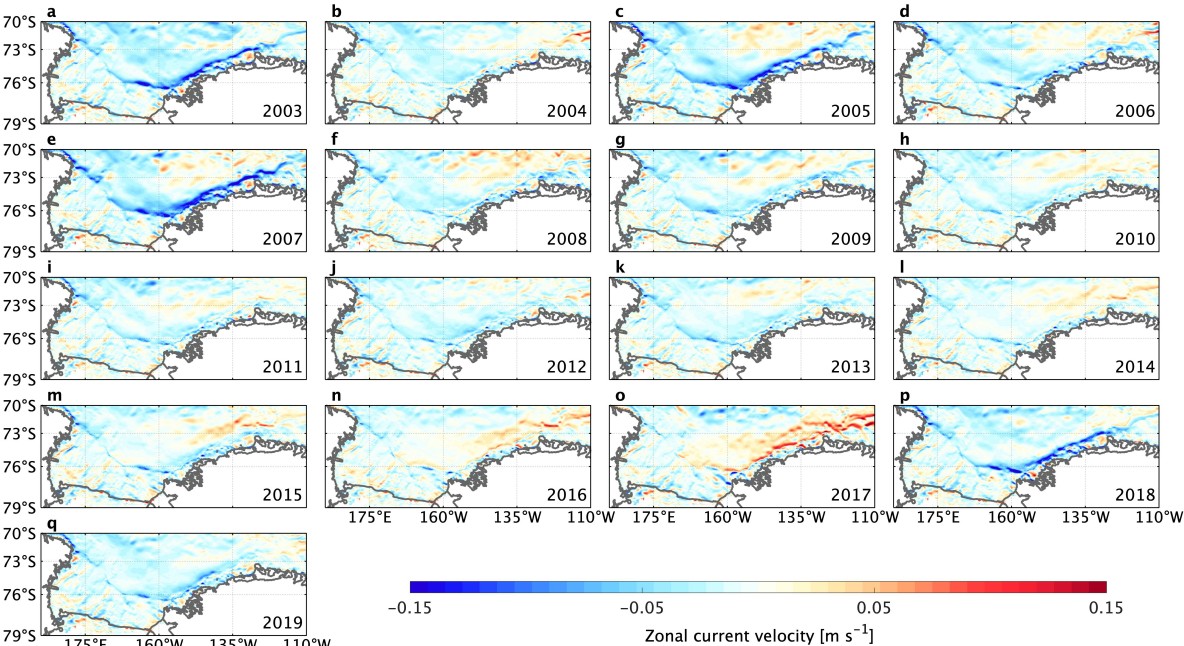

**Figure 9. (a–q)** Spatial distributions of winter-averaged zonal current velocities in the Ross Sea and Amundsen Sea from 2003 to 2019, averaged over the upper 300 m, where the majority of meltwater is located.

As described by Equation (1) in Section 2, freshwater flux is influenced not only by the amount of ice shelf meltwater but also by the magnitude of current velocity. To further investigate the cause of the extreme freshwater flux observed in winter 2007 (Fig. 8), we analyzed the spatial distribution of the winter-averaged zonal current velocity in the upper 300 meters in the Ross Sea and Amundsen Sea (Fig. 9). The results show that both the Antarctic slope current and the coastal current reached their strongest intensities in 2007 (Fig. 9e). This analysis suggests that the strong freshwater flux from the Amundsen Sea to the Ross Sea in 2007 was a combined result of the highest ice shelf melt rate and the strongest zonal flow velocity (Fig. 9e), resulting in anomalously low salinity in the Ross Sea in this year. Further examination of Fig. 9 indicates that relatively strong current velocities also occurred in 2003 and 2005 (Figs. 9a and 9c), and Fig. 8 shows correspondingly high freshwater fluxes in austral winter for these years. However, Fig. 7b reveals that the ice shelf melt rates during 2003 and 2005 were not particularly elevated. This suggests that in these years, the strong zonal current velocities likely played a dominant role in driving the westward freshwater flux, emphasizing the significant influence of current dynamics on freshwater transport.





**4 Discussion**

The variability and trends in the formation of HSSW and the hydrographic characteristics in the Ross Sea are influenced by large-scale climate modes or systems, including the Southern Annular Mode (SAM) and the Amundsen Sea Low (ASL) (Silvano et al., 2020; Zhang et al., 2024a). They can modulate local wind fields over the Ross Sea and Amundsen Sea, thereby affecting sea ice production in polynyas and the transport of freshwater from the Amundsen Sea to the Ross Sea. These processes ultimately affect the formation of HSSW, the hydrographic characteristics of the Ross Sea, and, subsequently, the stability of the meridional overturning circulation.

Projections from multiple CMIP5 models under the RCP8.5 scenario indicate that SAM is likely to continue shifting toward its positive phase by the end of this century (Zheng et al., 2013), which can strengthen the westerly jet and shift its core poleward, leading to a weakening of polar easterlies (Sen Gupta and England, 2006; Zhang et al., 2018). This weakening suppresses the westward transport of freshwater and sea ice from the Amundsen Sea driven by the Antarctic coastal current (Kim et al., 2016; Dotto et al., 2018; Silvano et al., 2020). Simultaneously, the weakening of polar easterlies can enhance offshore winds in the TNBP region and reduce surface air temperature over the RISP, which increases sea ice production and promotes greater HSSW/AABW production (Zhang et al., 2024a). The processes mentioned above can interact synergistically to enhance the HSSW formation in the future. In addition, previous research indicates that the positive phase of SAM is associated with a significant increase of cyclones near the Antarctic coast (Uotila et al., 2013; Grieger et al., 2018). Enhanced cyclonic activities will strengthen offshore winds over the western RISP, enlarging the polynya and promoting SIP (Wenta and Cassano, 2020; Wang et al., 2023). This further facilitates the HSSW formation and increases salinity in the Ross Sea. Therefore, by affecting both sea ice production in the Ross Sea polynyas and the freshwater transport from the Amundsen Sea, the future change of SAM is expected to drive a long-term increase in HSSW production and salinity in the Ross Sea.

Future projections based on CMIP5 and CMIP6 models indicate that under high-emission scenarios, the ASL will be deepened and shift poleward (Hosking et al., 2016; Gao et al., 2021). A deepened ASL may enhance offshore winds over the RISP and TNBP, increasing ice production in these polynyas and promoting subsequent HSSW production. Meanwhile, the poleward shift of the ASL is anticipated to weaken the easterly wind and the barotropic westward slope and coastal currents, reducing meltwater transport from the Amundsen Sea into the Ross Sea and further facilitating HSSW formation. Thus, both





the future trends of SAM and ASL may also support increased HSSW formation and salinity in the Ross
Sea. In the future, the Ross Sea may experience more frequent episodes of high ice production, intense
HSSW formation and increased salinity.

## 5 Conclusions

This study examines the mechanisms and impacts of HSSW formation in the Ross Sea, focusing on the
extreme event in 2007. Increased cyclonic activities over the Ross Ice Shelf in 2007 enhanced offshore
winds west of the Ross Ice Shelf polynya, leading to record-high sea ice production and HSSW formation.
However, salinity and density in the Ross Sea exhibited anomalously low values in this year. The
inconsistency between HSSW formation and salinity field was primarily driven by a significant influx of
ice shelf meltwater from the Amundsen Sea, contributed by both large ice shelf melting rate and strong
westward slope and coastal currents (Fig. 10). Analysis of seasonal freshwater fluxes revealed that the
strong westward transport of meltwater began in spring 2006, persisted through the summer between 2006
and 2007, and intensified into the autumn and winter of 2007. The time required for ice shelf meltwater to
travel from the western Amundsen Sea to the western Ross Sea and significantly alter local hydrographic
properties is approximately 9–12 months when the Antarctic slope and coastal currents are relatively
stronger. The peak freshwater flux during winter 2007, driven by increased ice shelf melting rates and
strong zonal current velocities, could play the most critical role in the unusually low salinity in the Ross
Sea in winter 2007. These findings highlight the role of synoptic atmospheric events on the accumulative
HSSW formation in winter, which can potentially make significant contributions to AABW production in
the open ocean. The impacts on AABW are not analyzed in this work but warrant further investigation in
the future. The results also highlight the complex interplay between sea ice production, meltwater fluxes
and ocean currents in shaping regional hydrographic characteristics in the Ross Sea, as well as the
importance of understanding the timing and persistence of freshwater fluxes and their downstream impacts
on hydrographic properties in the Ross Sea.





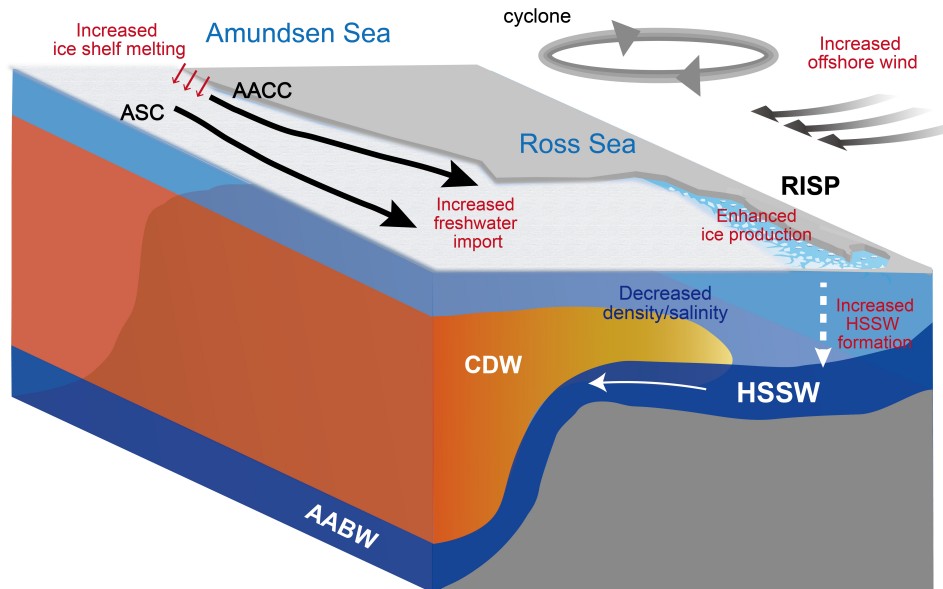

**Figure 10.** Schematic illustrating the atmospheric drivers of high salinity shelf water (HSSW) production in the Ross Sea and the combined effects of HSSW production and freshwater transport from the Amundsen Sea on the Ross Sea hydrographic fields. RISP represents the Ross Ice Shelf Polynya. CDW denotes the circumpolar deep water and AABW indicates the Antarctic Bottom Water. AACC represents the Antarctic coastal current and ASC refers to the Antarctic slope current.

Data availability.

The model data that support the findings of this study are available at https://doi.org/10.5072/zenodo.138444. More details about other observed and reanalysis data are presented in Sect. 2.

Author contributions.

ZZ and XW designed the original ideas presented in this manuscript. XW conducted the model results analysis. XW and ZZ wrote the original manuscript draft. CX, XZ, CW, HH and YC participated in the result interpretation, manuscript preparation and improvement. CX and HH conducted the 5-day-average model simulations. CX, HH CW and YC contributed to the model development. All authors contributed to the article and approved the submitted version.



Competing interests. The authors declare that there is no conflict of interest.

Acknowledgements.
This work is funded by the National Natural Science Foundation of China (Grant No. 42406259, No.
42476271, No. 41941008 and No. 42476257), the Shanghai Pilot Program for Basic Research of Shanghai
Jiao Tong University (Grant No. TQ1400201), the Impact and Response of Antarctic Seas to Climate
Change (Grant No IRASCC 1-02-01B), and the Shenlan Program funded by Shanghai Jiao Tong University
(Grant No. SL2020MS021).

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
