# Peer review of "Mechanisms and impacts of extreme high-salinity shelf water formation in the Ross Sea"

_EGUsphere, 2024_

## Author Response (AR2)

**Response to 2ⁿᵈ Review Comments**

for

**"Mechanisms and impacts of anomalous high-salinity shelf water formation in the Ross Sea"**

Xiaoqiao Wang, Zhaoru Zhang, Chuan Xie, Xi Zhao, Chuning Wang, Heng Hu, Yuanjie Chen

**Note**: Reviewers' comments are highlighted by blue color; authors' responses are in black color. Revisions in the revised manuscript are highlighted by blue color.

**Reviewer comments:**

**Anonymous Referee #2:**

I commend the authors for thoroughly addressing my previous concerns, significantly strengthening the manuscript. Before publication, I offer several concise suggestions for final polishing:

We thank the reviewer for his/her efforts in reviewing this manuscript and providing useful and helpful comments that further improved our manuscript.

Figure 2: Given the systematic bias in the mean state between simulations and observations (evident from differing y-axis ranges), would presenting the data as SIP anomalies instead of absolute values improve visual coherence?

Thank you for the insightful suggestion. Although a systematic bias between the simulations and observations is evident, Figure 2 still effectively illustrates a robust correlation between the two (R = 0.64, P = 0.04), despite differences in their absolute values. The overestimation of SIP by climate models has been widely reported in previous studies and is also addressed in our manuscript (Lines 218–225). Retaining the absolute values in Figure 2 is therefore intended to emphasize this commonly observed discrepancy, providing a clearer basis for the subsequent discussion on potential mechanisms underlying model–observation differences.

Figure 5: Please specify the unit of "Salinity" on the colorbar.

Figure 5 has been modified including the unit of salinity in the final version.

Lines 290 and 316: "between October and July" should be "between July and October"

This sentence has been revised in the final manuscript.

Figure 8: Please add a space between words and parentheses in the legend text.

The legend in Figure 8 has been revised according to the reviewer's suggestion.

Line 468: "alter surface heat loss and thus the ocean and atmosphere" is too general. Could you please rephrase this sentence to be more specific (i.e., the properties or processes of the ocean and atmosphere)?

According to the reviewer's suggestion, this sentence has been revised to "These dramatic changes in sea ice can substantially alter surface heat loss and thus water mass formation and cyclone systems".

Line 470: "increased atmospheric heat loss" should be "increased oceanic heat loss".

Sorry for this typo, the sentence has been revised to "oceanic heat loss".

Line 467-480: Aside from the two minor issues mentioned, I suggest the authors further polish the language in this paragraph. Specifically, the repetitive transitional phrases (e.g., the current structure alternates between "Furthermore... Overall... Furthermore... In summary") somewhat impede the logical flow. Could this sequence be simplified for better coherence?

Thanks for the reviewer's valuable comments. We have revised the paragraph accordingly, reducing the use of transitional phrases and improving the overall clarity and coherence of the text.

**Anonymous Referee #3:**

I thank the authors for revising the manuscript. It is now strongly improved. I have a few very minor comments that would not require a further review from my side.

We thank the reviewer for his/her efforts in reviewing this manuscript and providing useful and helpful comments that further improved our manuscript.

Ice shelf meltwater. I found this definition for water coming from the Amundsen Sea still not fully appropriate. What about "Amundsen Sea water"? Or something similar?

Thank you for the valuable suggestion. In the manuscript, the term "ice shelf meltwater" is intended to refer broadly to freshwater originating from both the Ross Ice Shelf and the Amundsen Sea ice shelves, rather than specifically to meltwater sourced solely from the Amundsen Sea. To further distinguish the contribution from the Amundsen Sea, we quantify the freshwater flux across the boundary section between the Amundsen and Ross seas. The use of the term meltwater is primarily to emphasize the low-salinity characteristics of this water mass. Therefore, we prefer to keep useing the term "ice shelf meltwater" in this context.

Observations are available both in 2007 (Jacobs et al. 2022; https://agupubs.onlinelibrary.wiley.com/doi/full/10.1029/2021JC017808) and summer 2008, reflecting dense water formation over 2007. So possibly a few more words about the comparison between model results and in situ measurements would be useful.

Thank you for this helpful comment. The primary focus of our study is on processes occurring during the austral winter of 2007. As noted in Jacobs et al. (2022), the observational data in that study were collected exclusively during the austral summer (DJF), specifically between 08 December and 25 February (see their Section 2). Given this seasonal difference, a direct comparison between our winter-focused model results and the summer in situ measurements is not straightforward. In light of this, we have not pursued a detailed comparison, as it may not be representative of the winter conditions of interest in our study.